# Bounds on Relativistic Deformed Kinematics from the Physics of the Universe Transparency

**José Manuel Carmona *** , **José Luis Cortés**, **Lucía Pereira** and **José Javier Relancio**

Departamento de Física Teórica and Centro de Astropartículas y Física de Altas Energías (CAPA), Universidad de Zaragoza, 50009 Zaragoza, Spain; cortes@unizar.es (J.L.C.); 706381@unizar.es (L.P.); relancio@unizar.es (J.J.R.)
* Correspondence: jcarmona@unizar.es

**Abstract:** We analyze the kinematics of electron-positron production in a photon-photon interaction when one has a modification of the special relativistic kinematics as a power expansion in the inverse of a new high-energy scale. We derive the equation for the threshold energy of this reaction to first order in this expansion, including the effects due to a modification of the energy-momentum conservation equation. In contrast with the Lorentz invariance violation case, a scale of the order of a few TeV is found to be compatible with the observations of very high-energy cosmic gamma rays in the case of a modification compatible with the relativity principle.

**Keywords:** quantum gravity; deformed kinematics; DSR; high-energy gamma-rays; pair production

## 1. Introduction

The detection of high-energy photons as cosmic messengers does not only provide information about the source that originated them, but also about the medium in which they propagate and the interactions they have suffered in their way to us. Because of this, their observation (or non-observation) offers the opportunity to test non-conventional physics that might alter the standard analysis of these processes.

Specifically, the flux of high-energy gamma rays suffers attenuation as a result of their interaction with the photons of the extragalactic background light (EBL), the radiation emitted by stars, galaxies, and active galactic nuclei since the reionization period, now present mainly in the optical and infrared bands (but also at lower wavelengths). Such interaction takes place for gamma rays with energies above the threshold of pair production, which could be altered by new physics.

In particular, quantum gravity models generically predict [1–15] deformations of the kinematics of special relativity in processes involving particles of sufficiently high energy, where this energy has to be compared with the high-energy scale $\Lambda$ that controls this deformation (disappearing in the limit $\Lambda \to \infty$), which is a parameter of the model. Such a deformation of the kinematics will generically alter the threshold of pair production, leading to a change in the expected gamma-ray flux, or an apparent failure in the estimate of the transparency of the universe to high-energy photons.

A modified dispersion relation with conventional conservation laws is the most prominent example of a deformed kinematics. It implies a violation of Lorentz invariance (LIV), since the deformed kinematics is defined and only valid in a specific set of reference frames related by rotations.

Modified dispersion relations in a LIV scenario are constrained by the time of flight of photons coming from gamma-ray bursts (GRBs) [16–18], active galactic nuclei [19], or pulsars [20]. In the framework of the standard model extension [21], a linear (proportional to $1/\Lambda$) variation of the speed of light also implies a birefringence effect, that may be tested in optical polarization measurements [22]. For a review on LIV phenomenology, see Reference [23].

Previous studies of LIV effects on the kinematics of electron–positron pair creation [24–27] have shown that indeed such effects shift the energy threshold of the pair-production process with respect to the case of special relativity, what may be used to constrain the scale $\Lambda$ of the LIV at the Planck-scale level. The effect can even be invoked as an explanation for anomalies in the absorption of high-energy gamma rays [28,29] (an apparent excess of transparency of the universe to them), although a LIV explanation of these anomalies is disfavored with respect to other type of new physics [30], mainly because the strong LIV needed would have been discarded by other observations, such as those involving atmospheric showers [31] or ultra-high energy cosmic rays [32].

However, LIV is not the only fate for a deformation of the kinematics of special relativity. Doubly special relativity models (DSR) emerged at the beginning of the century as quantum-gravity-inspired deformations of special relativity compatible with a relativity principle, that is, without a privileged system of reference, in which a length (the Planck scale) was observer-invariant [33–36]. DSR theories are an example of the general notion of a relativistic deformed kinematics (RDK), which does not only involve a modified dispersion relation, but also a modification of the energy-momentum composition rules that define the conservation laws [37–40]. This new ingredient, which is imposed by the existence of a relativity principle, makes the phenomenology of a RDK very different from that of a LIV scenario, invalidates many of the bounds for $\Lambda$ obtained in the LIV case, and leads to the possibility to have a high-energy scale for the deformed kinematics much smaller than the Planck scale [41,42].

In the present work we investigate the implications of a RDK on the threshold of electron-positron pair production, and, therefore, on the transparency of the universe to high-energy photons. The compatibility of experimental observations with these implications will allow us to put bounds on the high-energy scale $\Lambda$ of the RDK. As we will see, they are many orders of magnitude lower than in the case of LIV, which makes the RDK scenario much harder to exclude. This also indicates that the arguments disfavouring a deformation of special relativity as an explanation of possible anomalies in the transparency of the universe should be re-evaluated. On the other hand, it is interesting that these bounds are in the TeV regime, and could then be explored in other contexts, such as in future accelerator experiments. The transparency of the universe would therefore constitute a possible window to the TeV scale complementary to high-energy physics experiments.

The structure of the paper is as follows. In Section 2, we review the basics of a relativistic deformed kinematics and explain the main differences with respect to the LIV case. Then, in Section 3 we compare the calculation of the threshold of pair production for the special-relativistic and LIV cases with that of the RDK case, and obtain a relevant bound for the latter. Finally, in Section 4 we provide a discussion of the results. Detailed calculations indicated in the main text are given in the Appendix A.

## 2. Relativistic Deformed Kinematics

As commented in the Introduction, a deformed kinematics can either be compatible with a relativity principle (RDK), or represent a violation of the Lorentz invariance (LIV). While a deformed dispersion relation with standard conservation laws implies LIV, in a RDK there exists a second kinematic ingredient besides the deformed dispersion relation—a deformed composition law for the momenta, that needs to satisfy certain compatibility conditions with the deformed dispersion relation, known as "golden rules" [38,39]. Moreover, in order to maintain a relativity principle, deformed Lorentz transformations in the two-particle system are required [39,40].

While in both scenarios a deformed kinematics can be considered as a consequence of quantum gravity effects, its phenomenological implications are quite different. In LIV, the deformed dispersion relation is a way to take into account the propagation of a particle in a "quantum" spacetime; however, in DSR, besides this ingredient, there is also a lack of locality of interactions, known as relative locality [43]. These nonlocal effects are due to the deformed composition law—viewing the total momentum as the generator of translations, since the total momentum is a nonlinear function of the individual momenta, translations are different for each particle involved in the interaction, implying that only an observer placed at the interaction point sees the interaction as local. The lack of a notion

of absolute locality modifies completely the definition of space-time points thought by Einstein [44] as events given by the exchange of light pulses (emission and detection of photons). It is an open question whether this modified implementation of translations on a multi-particle system leads to observable effects in time-of-flight measurements [41,45,46]. If this is not the case then one has to look for effects of a RDK elsewhere.

Thresholds of reactions are also differently affected in a LIV or RDK scenario [42,47]. While thresholds in the decay of a particle cannot appear when going from SR kinematics to a relativistic deformed kinematics (the stability character of a particle cannot depend on its energy, since energy is not a relativistic invariant), they can indeed appear in a LIV theory. Also, thresholds of reactions are much more strongly affected in the case of LIV that in the case of RDK, as we will explicitly see in the present paper. As a consequence, the LIV and RDK scenarios lead to completely different bounds on the high-energy scale parametrizing the deviation from special relativity (SR).

To illustrate this, we will consider the simple case of an isotropic relativistic deformed kinematics at first order in an expansion in the inverse of the energy scale $\Lambda$ of the deformation, with a deformed dispersion relation parametrized by two dimensionless coefficients $\alpha_1, \alpha_2$:

$$C(p) \;=\; p_0^2 - \vec{p}^2 + \frac{\alpha_1}{\Lambda} p_0^3 + \frac{\alpha_2}{\Lambda} p_0 \vec{p}^2 = m^2\,, \tag{1}$$

and a deformed composition law parametrized by four adimensional coefficients $\beta_1, \beta_2, \gamma_1, \gamma_2$:

$$[p \oplus q]_0 \;=\; p_0 + q_0 + \frac{\beta_1}{\Lambda}\, p_0 q_0 + \frac{\beta_2}{\Lambda}\, \vec{p} \cdot \vec{q}, \quad [p \oplus q]_i \;=\; p_i + q_i + \frac{\gamma_1}{\Lambda}\, p_0 q_i + \frac{\gamma_2}{\Lambda}\, p_i q_0\,, \tag{2}$$

where the following condition is implemented

$$(p \oplus q)|_{q=0} \;=\; p\,, \qquad (p \oplus q)|_{p=0} \;=\; q\,. \tag{3}$$

This model was studied in Reference [39], where deformed Lorentz transformations laws were constructed. The relativity principle imposes the invariance of the deformed dispersion relation under the one-particle deformed Lorentz transformation $T(p)$,

$$C(p) = C(T(p))\,. \tag{4}$$

Non-linearity of the deformed composition law also forces to consider deformed Lorentz transformations in the two-particle system such that the transformation of the momentum of one particle depends on the momentum of the other particle, $T_q^L(p)$ and $T_p^R(q)$, where the superscripts $L, R$ indicate the relevance of the order in the composition of the momenta, since the composition law (2) is in general noncommutative. Imposing the relativity principle for a simple process (a particle with momentum $(p \oplus q)$ decaying into two particles of momenta $p$ and $q$) leads to

$$T(p \oplus q) \;=\; T_q^L(p) \oplus T_p^R(q)\,. \tag{5}$$

Equations (4) and (5) relate the deformed Lorentz transformations with the deformed dispersion relation and the deformed composition law. As a result (see Reference [39]), one obtains a relation between the dimensionless coefficients of the deformed dispersion relation and composition law, the "golden rules" we mentioned at the beginning of the section

$$\alpha_1 \;=\; -\beta_1\,, \qquad \alpha_2 \;=\; \gamma_1 + \gamma_2 - \beta_2\,. \tag{6}$$

As it was shown in Reference [40], there is a simple trick to obtain the previous relations without having to explicitly construct the deformed Lorentz transformations as it was done in Reference [39]. We can consider a *change of basis* from the momentum variables $P$ of SR, that is, a transformation in the

one-particle system $(P_0, \vec{P}) \to (p_0, \vec{p})$, where the new momentum variables are just a function of the old ones (preserving rotational invariance),

$$p_0 = P_0 + \frac{\delta_1}{\Lambda} P_0^2 + \frac{\delta_2}{\Lambda} \vec{P}^2,$$

$$p_i = P_i + \frac{\delta_3}{\Lambda} P_0 P_i. \tag{7}$$

This change of basis reproduces (at first order in the expansion in the inverse of the scale $\Lambda$) the terms in the dispersion relation and the coefficients of a symmetric composition law. In particular, we have

$$\alpha_1 = -2\delta_1, \quad \alpha_2 = -2\delta_2 + 2\delta_3, \quad \beta_1 = 2\delta_1, \tag{8}$$

$$\beta_2 = 2\delta_2, \quad \gamma_1 = \gamma_2 = \delta_3. \tag{9}$$

Moreover, we can also apply a *change of variables*, which is a transformation in the two-particle system which preserves the separation of momentum variables in the deformed dispersion relation. If $(P, Q)$ are variables that transform and compose linearly (the standard variables of SR), a change of variables $(P, Q) \to (p, q)$ with this property at order $1/\Lambda$, generating different coefficients $\gamma_1$ and $\gamma_2$ in the composition law of the variables $(p, q)$, is

$$P_0 = p_0 + \frac{\epsilon_1}{\Lambda} \vec{p} \cdot \vec{q}, \qquad P_i = p_i + \frac{\epsilon_1}{\Lambda} p_0 q_i,$$

$$Q_0 = q_0 + \frac{\epsilon_2}{\Lambda} \vec{p} \cdot \vec{q}, \qquad Q_i = q_i + \frac{\epsilon_2}{\Lambda} q_0 p_i. \tag{10}$$

Combining the change of variables with the change of basis, Equation (9) is replaced by

$$\beta_2 = 2\delta_2 + \epsilon_1 + \epsilon_2, \quad \gamma_1 = \delta_3 + \epsilon_1, \quad \gamma_2 = \delta_3 + \epsilon_2. \tag{11}$$

From Equations (8) and (11), we can directly derive the "golden rules" (6).

### 3. Threshold of Pair Production

In this section we are going to focus on the kinematics of electron-positron pair production, computing the threshold energy of the process under different kinematic considerations. That is, we want to find out the minimum energy of a high-energy photon which interacts with a low-energy photon belonging to the EBL to produce an electron-positron pair,

$$\gamma + \gamma_{\text{EBL}} \to e^- + e^+. \tag{12}$$

We will denote energy and momentum by $(E, \vec{k})$ for the high-energy photon and $(\varepsilon, \vec{k}')$ for the low-energy photon, leaving $(p_0, \vec{p})$ and $(q_0, \vec{q})$ for the electron and the positron, respectively.

First of all, we include a quick reminder (see Appendix A.1 for details) of the result obtained considering the dispersion relation and composition law of SR,

$$C(p) = p_0^2 - \vec{p}^2 = m^2, \tag{13}$$

$$[p \oplus q]_0 = p_0 + q_0, \qquad [p \oplus q]_i = p_i + q_i. \tag{14}$$

The threshold situation is reached when the momenta of all the particles are parallel, with $\vec{k}'$ pointing in the opposite direction to the other momenta, and $|\vec{q}| = |\vec{p}|$. Hence, the minimum energy for the high-energy photon takes the form

$$E_{\text{th}}^{\text{SR}} = \frac{m_e^2}{\varepsilon}. \tag{15}$$

We can now concentrate on a LIV situation with a deformed dispersion relation

$$C(p) = p_0^2 - \vec{p}^2 \left[ 1 - s\frac{p_0}{\Lambda} \right] = m^2. \tag{16}$$

The coefficient $s$ that appears in Equation (16) takes into account the possibility that a particle can travel faster ($s = -1$) or slower ($s = +1$) than their relativistic counterpart. This can result in a decrease or increase, respectively, of the threshold energy as we will see later in this section.

The threshold situation (see Appendix A.2 for details) is also reached when the momenta of all the particles are parallel with $\vec{k}'$ pointing in the opposite direction to the other momenta and with $|\vec{q}| = |\vec{p}|$ as in the case of special relativity kinematics. The modification of the dispersion relation leads to a modified equation for the threshold energy

$$-s\frac{E_{\text{th}}^3}{8\Lambda} + E_{\text{th}}\varepsilon - m_e^2 = 0, \tag{17}$$

where $s = -1$ would imply a decrease in the threshold energy with respect to the SR situation, and $s = +1$ corresponds to an increase in the threshold energy needed to produce the electron-positron pair, for a given energy $\varepsilon$ of the low-energy photon.

One can obtain an expression for the modification of the threshold if it is assumed that such modification is small, by substituting the special-relativistic threshold, $E_{\text{th}}^{\text{SR}} = m_e^2/\varepsilon$, in the term proportional to $1/\Lambda$ (this assumption will only hold for large enough values of $\Lambda$):

$$E_{\text{th}}^{\text{LIV}} \approx \frac{m_e^2}{\varepsilon} \left[ 1 + s\frac{(m_e^2)^2}{\varepsilon^3} \frac{1}{8\Lambda} \right]. \tag{18}$$

We can now proceed to discuss the case of a RDK scenario. Here we need to consider both a deformed dispersion relation and a deformed composition law (see Equations (1) and (2)) where the new coefficients $\alpha_i, \beta_i, \gamma_i$ that parametrize the deviations from SR are related to each other by means of the "golden rules" shown in Equation (6), so that the relativity principle is maintained.

One can generalize (see Appendix A.3 for details) the equation for the threshold energy including the effects due to a modification of the composition law of momenta. One then sees that, contrary to what happened in the SR and the LIV cases, the energies of the electron and positron at the threshold situation are not equal in the RDK case, owing to a non-symmetric ($\gamma_1 \neq \gamma_2$) deformed composition law. When one uses the same approximations as in the case of LIV, one finds

$$\frac{\gamma_1 + \gamma_2 - \beta_1 - \beta_2 - \alpha_1 - \alpha_2}{8\Lambda} E_{\text{th}}^3 + E_{\text{th}}\varepsilon - m_e^2 = 0. \tag{19}$$

This correction shows a cubic equation for the threshold energy, the same order obtained in Equation (17) for a LIV situation. In fact, the generalized equation for the threshold energy Equation (19) reduces to Equation (17) in the case of LIV ($\gamma_i = \beta_i = 0$) with a redefinition of the energy scale $\Lambda$, such that $(\alpha_1 + \alpha_2)/\Lambda \to s/\Lambda$. However, when the coefficients of the deformed dispersion relation and composition law in a RDK are forced to fulfill the "golden rules" of Equation (6) by the relativity principle, one has a cancellation of the contribution proportional to $E_{\text{th}}^3$ in Equation (19) for the threshold energy.

We then look for the first corrections proportional to $1/\Lambda$ where the contributions from the different terms do not cancel when considering the "golden rules", giving a quadratic equation,

$$\frac{3\gamma_1 + \gamma_2 - \beta_1 - 5\beta_2}{4\Lambda} E_{\text{th}}^2 \varepsilon + \frac{2\beta_2 - \gamma_1 - 2\gamma_2}{2\Lambda} E_{\text{th}} m_e^2 + E_{\text{th}}\varepsilon - m_e^2 = 0. \tag{20}$$

We can now substitute the relativistic solution for $E_{\text{th}}$ given by Equation (15) in the terms proportional to $1/\Lambda$. The result for the threshold energy in the context of a relativistic deformed kinematics is

$$E_{\text{th}}^{\text{RDK}} \approx \frac{m_e^2}{\varepsilon}\left[1 + \frac{\beta_1 + \beta_2 + 3\gamma_2 - \gamma_1}{4\Lambda}\frac{m_e^2}{\varepsilon}\right] = \frac{m_e^2}{\varepsilon}\left[1 + \frac{m_e^2}{\varepsilon\Lambda_{\text{eff}}}\right], \tag{21}$$

where $\Lambda_{\text{eff}}$ is the effective deformation scale for pair-production, defined as a function of the high-energy scale $\Lambda$ and the parameters $\beta_1, \beta_2, \gamma_1$, and $\gamma_2$. Comparison with Equation (18) shows the difference with the modification of the threshold in the LIV case by the large factor $(m_e^2)/\varepsilon^2$. The alteration of the kinematics of SR is then substantially different in the LIV and the RDK cases.

The approximations used throughout this section are based on the hypothesis that the modification of the threshold energy due to RDK is much smaller than the threshold energy in special relativity, $|E_{\text{th}}^{\text{RDK}} - E_{\text{th}}^{\text{SR}}| \ll E_{\text{th}}^{\text{SR}}$. We can quantify this by considering that their difference is, for example,

$$\frac{E_{\text{th}}^{\text{RDK}} - E_{\text{th}}^{\text{SR}}}{E_{\text{th}}^{\text{SR}}} < 0.1. \tag{22}$$

Then, the previous equation will give $E_{\text{th}}^{\text{RDK}} < 1.1 E_{\text{th}}^{\text{SR}}$. From Equations (15) and (21), we obtain a bound for the effective QG modification scale, $\Lambda_{\text{eff}}$,

$$\Lambda_{\text{eff}} > \frac{10 m_e^2}{\varepsilon}. \tag{23}$$

If we take, for example, an EBL photon of wavelenght $\lambda = 1\,\mu\text{m}$ and energy $\varepsilon = 1.24\,\text{eV}$, the effective scale would take a value of $\Lambda_{\text{eff}} > 2.1\,\text{TeV}$. Hence, we can infer the order of the modification scale knowing the characteristics of the low-energy photon.

## 4. Conclusions and Outlook

We have applied a general modification of special relativistic kinematics, proportional to the inverse of a new energy scale $\Lambda$, to the determination of the threshold of the production of an electron-positron pair in the interaction of a high-energy ($E$) photon with a low-energy ($\varepsilon$) photon in the extra-galactic background. In the general case, one finds corrections proportional to the ratio $(E^3/m_e^2\Lambda)$ in a cubic equation for the threshold energy so that one can have large corrections to the threshold energy even when $\Lambda \gg E$. This situation is the one commonly discussed in the literature. However, when the modification of the kinematics includes terms proportional to the inverse of the scale $\Lambda$ in the composition law of momenta, such that the deformed kinematics is compatible with the relativity principle, one finds that the dominant correction term in the equation of the threshold energy is absent and the correction turns out to be proportional to the ratio $(E/\Lambda)$. An upper bound on a possible deviation of the threshold energy from the result derived with SR kinematics can then be used to put a lower bound on the scale $\Lambda$. Interestingly, this bound (TeV scale) is many orders of magnitude lower than in the more conventional case of a Lorentz invariance violation.

An analysis based on the modification of the threshold energy in the electron-positron pair production to consider the problem of the transparency of the Universe to high-energy gamma rays is of course incomplete and can only give qualitative indications. A more detailed analysis, taking into account specific models for the EBL and a determination of the optical depth, as in References [19,26], should be performed. Such study will require to go beyond the determination of the threshold energy considering all the effects of the deformation of the kinematics in the determination of the VHE gamma-ray spectrum. One could even consider a situation where the accuracy in the determination of the gamma-ray spectrum requires to go beyond the terms proportional to the inverse of the scale $\Lambda$ in the modification of the kinematics. There is at present a wealth of data from H.E.S.S., HAWC or MAGIC, where this analysis could be carried out, and we will have more data in the near future with CTA. Such analysis can be used to get stringent bounds on the scale $\Lambda$ for a relativistic deformed

kinematics if one does not find a conflict with special relativistic kinematics. Alternatively, one could find a spectrum which is not compatible with the predictions based on SR kinematics, and one should see whether the incompatibility could be adjusted with an appropriate choice of the scale $\Lambda$.

In this sense, the discussion of the gamma-ray spectrum in relation to the transparency of the Universe presented in this work should be considered together with other observations which can also be affected by a deformation of SR kinematics, including the end of the UHECR spectrum, observations of cosmogenic neutrinos, and high-energy collider physics (see Reference [48] as an example), where new data are also expected in the near future. Consistency with these other observations will also indirectly contribute to a better knowledge about the physics of the transparency of the Universe to gamma rays, by constraining the role of new physics in the origin of possible anomalies in the detected gamma-ray spectrum or tracing them down to a lack of understanding of the EBL spectrum. The low-energy bounds obtained in the present work for a relativistic deformed kinematics makes this a promising approach with important astrophysical consequences.

**Author Contributions:** Writing—original draft, J.M.C., J.L.C., L.P. and J.J.R. All authors contributed equally to the present work. All authors have read and agreed to the published version of the manuscript.

**Funding:** This work is supported by the Spanish grants PGC2018-095328-B-I00 (FEDER/Agencia estatal de investigación), and DGAFSE grant E21-20R. The authors would like to acknowledge the contribution of the COST Action CA18108.

**Conflicts of Interest:** The authors declare no conflict of interest.

## Appendix A. Equation for the Threshold Energy of Pair Production

In this Appendix we display some details regarding the derivation of the expression for the threshold energy of pair production in different cases: SR kinematics, a LIV scenario with a modification of the dispersion relation, and a relativistic deformed kinematics with modifications in the composition of momenta and in the dispersion relation compatible with the relativity principle.

*Appendix A.1. Threshold in SR*

In the first place, we compute the result obtained for the minimum energy of the high-energy photon in the process according to special relativity kinematics. The dispersion relation and the composition law are

$$C(p) = p_0^2 - \vec{p}^2 = m^2, \tag{A1}$$

$$[p \oplus q]_0 = p_0 + q_0, \qquad [p \oplus q]_i = p_i + q_i. \tag{A2}$$

In special relativity, the following quantity is found to be invariant for different inertial observers (i.e., under Lorentz transformations):

$$s = E_{\text{tot}}^2 - |\vec{p}_{\text{tot}}|^2, \tag{A3}$$

which will be useful to solve the problem by considering different frames of reference. We begin with the center of mass reference frame for the pair produced in the process, where the momenta of each of the two particles is zero in the threshold situation,

$$s_{\text{f}} = 4m_e^2. \tag{A4}$$

On the other hand, we can compute the previous invariant for the initial state in the laboratory frame, using Equation (A2) to calculate the total energy and momentum of the two-photon system,

$$s_{\text{i}} = \left(E_{\text{i}}\right)^2 - |\vec{p}_{\text{i}}|^2 = (E + \varepsilon)^2 - \left|\vec{k} + \vec{k}'\right|^2. \tag{A5}$$

Since the relativistic invariant *s* is conserved in the interaction, we can now equate Equations (A4) and (A5), simplifying the expression that remains by the use of the dispersion relation Equation (A1),

$$2E\varepsilon - 2\vec{k}\cdot\vec{k'} = 4m_e^2. \tag{A6}$$

Therefore, the energy of the high-energy photon in this situation depends on the angle $\theta$ between the two initial momenta of the photons,

$$E = \frac{2m_e^2}{\varepsilon(1-\cos\theta)}. \tag{A7}$$

Finally, the minimum energy corresponding to the threshold of the process is obtained when the two initial momenta are pointing in opposite directions ($\theta = \pi$),

$$E_{\text{th}}^{\text{SR}} = \frac{m_e^2}{\varepsilon}. \tag{A8}$$

*Appendix A.2. Threshold Equation with LIV*

In a scenario with Lorentz invariance violation, there is a modification in the dispersion relation of a particle (here we consider it only up to first order in $1/\Lambda$), while the composition law remains as in special relativity,

$$C(p) = p_0^2 - \vec{p}^2\left[1 - s\frac{p_0}{\Lambda}\right] = m^2, \tag{A9}$$

$$[p\oplus q]_0 = p_0 + q_0, \qquad [p\oplus q]_i = p_i + q_i. \tag{A10}$$

It is important to note that we are facing an optimization problem, as we are searching for the minimum possible energy of the high-energy photon. Therefore, it can be solved using the methodology of Lagrange multipliers, looking for the minimization of *E*, subject to the constraints given by the conservation laws of energy and momenta. The auxiliary function we use for that is

$$F(\vec{k},\vec{p},\vec{q},\mu,\vec{\lambda}) = E - \mu\left[p_0 + q_0 - E - \varepsilon\right] - \sum_i \lambda_i\left[p_i + q_i - k_i - k_i'\right], \tag{A11}$$

where $\mu$ and $\lambda_i$ are the so-called Lagrange multipliers. The next step in the optimization method would be to compute the derivatives of the new function *F* so that its minimum can be found, $\nabla \cdot F = 0$.

One has

$$\frac{\partial F}{\partial p_i} = -\mu\frac{\mathrm{d}p_0}{\mathrm{d}|\vec{p}|}\cdot\frac{p_i}{|\vec{p}|} - \lambda_i = 0, \tag{A12}$$

$$\frac{\partial F}{\partial q_i} = -\mu\frac{\mathrm{d}q_0}{\mathrm{d}|\vec{q}|}\cdot\frac{q_i}{|\vec{q}|} - \lambda_i = 0, \tag{A13}$$

$$\frac{\partial F}{\partial k_i} = (1+\mu)\frac{\mathrm{d}E}{\mathrm{d}|\vec{k}|}\cdot\frac{k_i}{|\vec{k}|} + \lambda_i = 0, \tag{A14}$$

$$\frac{\partial F}{\partial \mu} = p_0 + q_0 - E - \varepsilon = 0, \tag{A15}$$

$$\frac{\partial F}{\partial \lambda_i} = p_i + q_i - k_i - k_i' = 0. \tag{A16}$$

Matching the expressions for the multiplier $\lambda_i$ obtained from Equations (A12)–(A14), we get

$$\mu\frac{\mathrm{d}p_0}{\mathrm{d}|\vec{p}|}\cdot\frac{p_i}{|\vec{p}|} = \mu\frac{\mathrm{d}q_0}{\mathrm{d}|\vec{q}|}\cdot\frac{q_i}{|\vec{q}|} = (1+\mu)\frac{\mathrm{d}E}{\mathrm{d}|\vec{k}|}\cdot\frac{k_i}{|\vec{k}|}. \tag{A17}$$

This equation shows that the unit vectors defining the directions of the momenta involved in the pair production process are proportional to each other. Hence, the problem at hand can be simplified to one dimension.

The first equality in Equation (A17) shows that the electron and the positron move in the same direction ($p_i/|\vec{p}|$) and with the same velocity ($\mathrm{d}p_0/\mathrm{d}|\vec{p}|$), so that they both have the same momentum and, consequently, the same energy. Indeed, we can obtain the velocity of each particle from the dispersion relation, Equation (A9), with $m = m_e$. Taking into account that we are working up to order $1/\Lambda$, the term $\vec{p}^2$ can be substituted by its relativistic expression in the $1/\Lambda$ term:

$$p_0^2 \approx |\vec{p}|^2 - s\left(p_0^2 - m_e^2\right)\frac{p_0}{\Lambda} + m_e^2. \tag{A18}$$

Since we are considering the ultra-relativistic limit $m_e \ll p_0 \ll \Lambda$, we get (neglecting terms proportional to $m_e^{2n}$ with $n > 1$)

$$|\vec{p}| \approx p_0 + s\frac{p_0^2}{2\Lambda} - s\frac{m_e^2}{2\Lambda} - \frac{m_e^2}{2p_0}, \tag{A19}$$

so that

$$\frac{\mathrm{d}p_0}{\mathrm{d}|\vec{p}|} = \frac{1}{\mathrm{d}|\vec{p}|/\mathrm{d}p_0} = \frac{1}{1 + sp_0/\Lambda + m_e^2/2p_0^2} \approx 1 - s\frac{p_0}{\Lambda} - \frac{m_e^2}{2p_0^2}. \tag{A20}$$

Then, the equality of velocities $\mathrm{d}p_0/\mathrm{d}|\vec{p}| = \mathrm{d}q_0/\mathrm{d}|\vec{q}|$ in Equation (A17) becomes

$$1 - s\frac{p_0}{\Lambda} - \frac{m_e^2}{2p_0^2} = 1 - s\frac{q_0}{\Lambda} - \frac{m_e^2}{2q_0^2}, \tag{A21}$$

$$|\vec{p}| = |\vec{q}|, \qquad p_0 = q_0. \tag{A22}$$

Considering the second equality in Equation (A17), we can clear the Lagrange multiplier $\mu$. We then observe that it will always take a positive value with $|\mu| < 1$, as $m_e \ll E \ll \Lambda$,

$$\mu\frac{\mathrm{d}q_0}{\mathrm{d}|\vec{q}|} = (1+\mu)\frac{\mathrm{d}E}{\mathrm{d}|\vec{k}|}, \qquad \mu\left(1 - s\frac{q_0}{\Lambda} - \frac{m_e^2}{2q_0^2}\right) = (1+\mu)\left(1 - s\frac{E}{\Lambda}\right), \tag{A23}$$

$$\mu = \frac{1 - sE/\Lambda}{s(E - q_0)/\Lambda - m_e^2/2q_0^2}. \tag{A24}$$

This result shows that the initial high-energy photon momenta $\vec{k}$ points in the same direction as the momenta $\vec{p}$ and $\vec{q}$ of the electron and positron. With this knowledge, we can now write the conservation of energy and momentum (Equations (A15) and (A16)) as

$$2p_0 = E + \varepsilon, \tag{A25}$$

$$2|\vec{p}| = |\vec{k}| \pm |\vec{k'}|. \tag{A26}$$

It is easy to note that the scenario leading to the minimum energy $E$ corresponds to the minus sign in Equation (A26), since in this case $|\vec{p}|$, and then $p_0$ in Equation (A25), takes its minimum value. Equations (A25) and (A26) are in fact the same as in SR, where we already saw that the threshold situation was at $\theta = \pi$, that is, when $\vec{k'}$ is opposite to $\vec{k}$, $\vec{p}$ and $\vec{q}$, so that

$$2|\vec{p}| = |\vec{k}| - |\vec{k'}|. \tag{A27}$$

Using now the dispersion relation (A9), or better, its approximation (A19), for the different particles, into Equation (A27), and substituting $p_0$ by using Equation (A25), we finally find an

expression for the energy $E$ of the high-energy photon in terms of the energy $\varepsilon$ of the low-energy photon. Taking into account that $\varepsilon \ll E$, the equation for the threshold energy in the LIV case is

$$-s\frac{E_{\text{th}}^3}{8\Lambda} + E_{\text{th}}\varepsilon - m_e^2 = 0, \tag{A28}$$

where $s = +1$ corresponds to the high-energy photon being subluminal, and $s = -1$ relates to the superluminal situation.

*Appendix A.3. Threshold Equation with a RDK*

The last set of calculations presented in this appendix refers to a relativistic deformed kinematics situation. While this scenario maintains the relativity principle, it is defined by a deformed dispersion relation and a deformed composition law,

$$C(p) = p_0^2 - \vec{p}^2 + \frac{\alpha_1}{\Lambda}p_0^3 + \frac{\alpha_2}{\Lambda}p_0\vec{p}^2 = m^2, \tag{A29}$$

$$[p \oplus q]_0 = p_0 + q_0 + \frac{\beta_1}{\Lambda}p_0q_0 + \frac{\beta_2}{\Lambda}\vec{p}\cdot\vec{q}, \quad [p \oplus q]_i = p_i + q_i + \frac{\gamma_1}{\Lambda}p_0q_i + \frac{\gamma_2}{\Lambda}p_iq_0, \tag{A30}$$

whose dimensionless coefficients are related by means of the "golden rules" (6),

$$\alpha_1 = -\beta_1, \qquad \alpha_2 = \gamma_1 + \gamma_2 - \beta_2. \tag{A31}$$

It can be noticed that the deformed composition law for momenta does not take the same expression under an exchange of the involved momenta, $[p \oplus q]_i \neq [q \oplus p]_i$, if $\gamma_1 \neq \gamma_2$, so we must specify the composition order when applying Equation (A30).

As in the previous case, we will follow the Lagrange multipliers method in order to find the minimum energy $E$ for the process to occur. We define the auxiliary function $F$ as

$$F(\vec{k}, \vec{p}, \vec{q}, \mu, \vec{\lambda}) = E - \mu\left(E_{\text{fin}} - E_{\text{ini}}\right) - \sum_i \lambda_i\left[(p_{\text{fin}})_i - (p_{\text{ini}})_i\right], \tag{A32}$$

which allows one to minimize $E$ under the constraints given by energy and momentum conservation in the pair production process.

The initial and final expressions for the total energy and momenta, according to the composition law (A30), are the following:

$$E_{\text{ini}} = E + \varepsilon + \frac{\beta_1}{\Lambda}E\varepsilon + \frac{\beta_2}{\Lambda}\sum_i k_i k_i', \tag{A33}$$

$$E_{\text{fin}} = p_0 + q_0 + \frac{\beta_1}{\Lambda}p_0q_0 + \frac{\beta_2}{\Lambda}\sum_i p_i q_i, \tag{A34}$$

$$(p_{\text{ini}})_i = k_i + k_i' + \frac{\gamma_1}{\Lambda}Ek_i' + \frac{\gamma_2}{\Lambda}\varepsilon k_i, \tag{A35}$$

$$(p_{\text{fin}})_i = p_i + q_i + \frac{\gamma_1}{\Lambda}p_0q_i + \frac{\gamma_2}{\Lambda}q_0p_i. \tag{A36}$$

The next step in the Lagrange multipliers method, is to cancel the different derivatives obtained from the auxiliary function $F$:

$$\frac{\partial F}{\partial p_i} = -\mu \frac{\partial E_{\text{fin}}}{\partial p_i} - \lambda_i \frac{\partial (p_{\text{fin}})_i}{\partial p_i} = 0, \tag{A37}$$

$$\frac{\partial F}{\partial q_i} = -\mu \frac{\partial E_{\text{fin}}}{\partial q_i} - \lambda_i \frac{\partial (p_{\text{fin}})_i}{\partial q_i} = 0, \tag{A38}$$

$$\frac{\partial F}{\partial k_i} = \frac{\mathrm{d}E}{\mathrm{d}k_i} + \mu \frac{\mathrm{d}E_{\text{ini}}}{\mathrm{d}k_i} + \lambda_i \frac{\mathrm{d}(p_{\text{ini}})_i}{\mathrm{d}k_l} = 0, \tag{A39}$$

$$\frac{\partial F}{\partial \mu} = E_{\text{fin}} - E_{\text{ini}} = 0, \tag{A40}$$

$$\frac{\partial F}{\partial \lambda_i} = (p_{\text{fin}})_i - (p_{\text{ini}})_i = 0. \tag{A41}$$

The explicit expressions of Equations (A37)–(A39) are

$$\frac{\partial F}{\partial p_i} = -\mu \left[ \left( 1 + \frac{\beta_1}{\Lambda} q_0 \right) \frac{\mathrm{d}p_0}{\mathrm{d}|\vec{p}|} + \frac{\beta_2}{\Lambda} q_i \frac{|\vec{p}|}{p_i} \right] \frac{p_i}{|\vec{p}|} - \lambda_i \left[ 1 + \frac{\gamma_2}{\Lambda} q_0 + \frac{\gamma_1}{\Lambda} q_i \frac{\mathrm{d}p_0}{\mathrm{d}p_i} \right] = 0, \tag{A42}$$

$$\frac{\partial F}{\partial q_i} = -\mu \left[ \left( 1 + \frac{\beta_1}{\Lambda} p_0 \right) \frac{\mathrm{d}q_0}{\mathrm{d}|\vec{q}|} + \frac{\beta_2}{\Lambda} p_i \frac{|\vec{q}|}{q_i} \right] \frac{q_i}{|\vec{q}|} - \lambda_i \left[ 1 + \frac{\gamma_1}{\Lambda} p_0 + \frac{\gamma_2}{\Lambda} p_i \frac{\mathrm{d}q_0}{\mathrm{d}q_i} \right] = 0, \tag{A43}$$

$$\frac{\partial F}{\partial k_i} = \left[ \left( 1 + \mu + \mu \frac{\beta_1}{\Lambda} \varepsilon \right) \frac{\mathrm{d}E}{\mathrm{d}|\vec{k}|} + \mu \frac{\beta_2}{\Lambda} k_i' \frac{|\vec{k}|}{k_i} \right] \frac{k_i}{|\vec{k}|} + \lambda_i \left[ 1 + \frac{\gamma_2}{\Lambda} \varepsilon + \frac{\gamma_1}{\Lambda} k_i' \frac{\mathrm{d}E}{\mathrm{d}k_i} \right] = 0. \tag{A44}$$

The previous equations indicate that the unit vectors $p_i/|\vec{p}|$, $q_i/|\vec{q}|$ and $k_i/|\vec{k}|$ are proportional to each other, that is, the momenta of the high-energy photon and of the electron-positron pair are parallel. From momentum conservation, the momentum of the low-energy photon will also share the same direction and, therefore, the problem can be reduced to one dimension, the direction of the momenta. Moreover, the zeroth-order correction in $1/\Lambda$, that is, the case of special relativity, indicates that the momentum of the low-energy photon points in opposite direction to the other momenta involved in the pair-production, so that conservation of energy and momentum can be written as:

$$E + \varepsilon + \frac{\beta_1}{\Lambda} E \varepsilon - \frac{\beta_2}{\Lambda} k \cdot k' = p_0 + q_0 + \frac{\beta_1}{\Lambda} p_0 q_0 + \frac{\beta_2}{\Lambda} p \cdot q, \tag{A45}$$

$$k - k' - \frac{\gamma_1}{\Lambda} E k' + \frac{\gamma_2}{\Lambda} \varepsilon k = p + q + \frac{\gamma_1}{\Lambda} p_0 q + \frac{\gamma_2}{\Lambda} q_0 p. \tag{A46}$$

As we did in the previous section, the dispersion relation (A29) can be written, in the ultrarelativistic limit $m_e \ll p_0 \ll \Lambda$, as

$$|\vec{p}| \approx p_0 + \frac{\alpha_1 + \alpha_2}{2\Lambda} p_0^2 - \frac{\alpha_2}{2\Lambda} m_e^2 - \frac{m_e^2}{2p_0}. \tag{A47}$$

Using this and Equations (A42) and (A43), we get that, in opposition to what happened in the SR and LIV cases, the energies of the electron and positron are not the same, but they are related by

$$q_0 \approx p_0 + \frac{\gamma_1 - \gamma_2}{\Lambda} p_0^2 + \frac{3(\gamma_2 - \gamma_1)}{4\Lambda} m_e^2. \tag{A48}$$

Writing in Equations (A45) and (A46) the momenta of the particles as a function of their energies [Equation (A47)] and the expression of $q_0$ in terms of $p_0$ [Equation (A48)], one gets

$$\frac{\beta_1 + \beta_2 + \gamma_1 - \gamma_2}{\Lambda} p_0^2 + 2p_0 + \frac{3\gamma_2 - 3\gamma_1 - 4\beta_2}{4\Lambda} m_e^2 = E + \varepsilon + \frac{\beta_1 - \beta_2}{\Lambda} E \varepsilon, \tag{A49}$$

$$\frac{\alpha_1 + \alpha_2 + 2\gamma_1}{\Lambda} p_0^2 + 2p_0 - \frac{m_e^2}{p_0} - \frac{\gamma_2 + 3\gamma_1 + 4\alpha_2}{4\Lambda} m_e^2 = E - \varepsilon + \frac{\alpha_1 + \alpha_2}{2\Lambda} E^2 + \frac{\gamma_2 - \gamma_1}{\Lambda} E \varepsilon. \tag{A50}$$

We can also approximate $p_0$ in the terms proportional to $1/\Lambda$ to the SR result, $(E + \varepsilon)/2$. Then we have:

$$\frac{\beta_1 + \beta_2 + \gamma_1 - \gamma_2}{4\Lambda} (E + \varepsilon)^2 + 2p_0 + \frac{3\gamma_2 - 3\gamma_1 - 4\beta_2}{4\Lambda} m_e^2 =$$

$$= E + \varepsilon + \frac{\beta_1 - \beta_2}{\Lambda} E\varepsilon, \tag{A51}$$

$$\frac{\alpha_1 + \alpha_2 + 2\gamma_1}{4\Lambda} (E + \varepsilon)^2 + 2p_0 - \frac{m_e^2}{p_0} - \frac{\gamma_2 + 3\gamma_1 + 4\alpha_2}{4\Lambda} m_e^2 =$$

$$= E - \varepsilon + \frac{\alpha_1 + \alpha_2}{2\Lambda} E^2 + \frac{\gamma_2 - \gamma_1}{\Lambda} E\varepsilon. \tag{A52}$$

From Equation (A51), we obtain the energy $p_0$ of the electron as a function of the energy $E$ of the high-energy photon up to first order in $1/\Lambda$,

$$p_0 \approx \frac{E + \varepsilon}{2} + \frac{\beta_1 - \beta_2}{2\Lambda} E\varepsilon + \frac{3\gamma_1 - 3\gamma_2 + 4\beta_2}{8\Lambda} m_e^2 - \frac{\beta_1 + \beta_2 + \gamma_1 - \gamma_2}{8\Lambda} (E + \varepsilon)^2. \tag{A53}$$

Using the previous result into Equation (A52), we obtain an equation for the first correction to the threshold energy for the high-energy photon:

$$\frac{\gamma_1 + \gamma_2 - \beta_1 - \beta_2 - \alpha_1 - \alpha_2}{8\Lambda} E^3 + E\varepsilon - m_e^2 = 0. \tag{A54}$$

Now, we note that the first term in the previous equation is automatically cancelled, since the "golden rules" (A31) must be satisfied in a relativistic theory. This means that in a RDK theory, the first correction to the threshold energy $E$ will not be given by a cubic equation, as it was the case in the LIV scenario.

Therefore, the threshold energy in a relativistic deformed kinematics suffers a lower modification from the SR result than in a LIV scenario. Its expression can be found by considering the next correction (proportional to $\varepsilon/\Lambda$ or to $m_e^2/\Lambda$) in Equation (A52),

$$\frac{3\gamma_1 + \gamma_2 - \beta_1 - 5\beta_2}{4\Lambda} E^2\varepsilon + \frac{2\beta_2 - \gamma_1 - 2\gamma_2}{2\Lambda} Em_e^2 + E\varepsilon - m_e^2 = 0. \tag{A55}$$

We can now replace $E$ by its relativistic result $(m_e^2/\varepsilon)$ in the terms inversely proportional to the quantum gravity scale $\Lambda$. Then one finds:

$$E_{\text{th}}^{\text{RDK}} \approx \frac{m_e^2}{\varepsilon} \left[ 1 + \frac{\beta_1 + \beta_2 + 3\gamma_2 - \gamma_1}{4\Lambda} \frac{m_e^2}{\varepsilon} \right] = \frac{m_e^2}{\varepsilon} \left[ 1 + \frac{m_e^2}{\varepsilon\Lambda_{\text{eff}}} \right]. \tag{A56}$$

The coefficients of the deformed composition law and the modification scale $\Lambda$ define an effective deformation scale, $\Lambda_{\text{eff}} = (\beta_1 + \beta_2 + 3\gamma_2 - \gamma_1)/4\Lambda$, which can be useful to estimate the threshold energy difference with respect to the SR result.

As indicated previously, the order of the composition of momenta is relevant in a RDK scenario. Had we considered the combination $[q \oplus p]_i$ instead of $[p \oplus q]_i$ for the electron-positron pair, the coefficients $\gamma_1$ and $\gamma_2$ would be exchanged in the deformed composition law, and also in the final expression obtained for the threshold energy:

$$E_{\text{th}}^{\text{RDK}'} \approx \frac{m_e^2}{\varepsilon} \left[ 1 + \frac{\beta_1 + \beta_2 + 3\gamma_1 - \gamma_2}{4\Lambda} \frac{m_e^2}{\varepsilon} \right] = \frac{m_e^2}{\varepsilon} \left[ 1 + \frac{m_e^2}{\varepsilon\Lambda_{\text{eff}}'} \right]. \tag{A57}$$

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
