# Peer review of "Bounds on Relativistic Deformed Kinematics from the Physics of the Universe Transparency"

_symmetry, doi:10.3390/sym12081298_

Round 1
Reviewer 1 Report
In the manuscript the authors analyze the threshold energies for the electron-positron pair creation in the framework of relativistic kinematics with modified dispersion laws. The energy corrections are obtained in terms of a fundamental energy scale of the deformation. The details of the calculations are presented in several Appendices. Even that in the title the authors claim to discuss "Universe transparency", very little is mentioned in the text about the cosmological or astrophysical applications of the obtained results. This manuscript may be publishable in Symmetry if the authors would present some concrete cosmological/astrophysical applications that would explicitly show how the present results are related to the "Universe transparency".
Author Response
We thank the referee for his/her comments and try to answer here to his/her main objection:
"Even that in the title the authors claim to discuss "Universe transparency", very little is mentioned in the text about the cosmological or astrophysical applications of the obtained results. This manuscript may be publishable in Symmetry if the authors would present some concrete cosmological/astrophysical applications that would explicitly show how the present results are related to the "Universe transparency"."
The process of electron-positron pair creation by the interaction of high-energy gamma rays with the low-energy photons of the extragalactic background light defines the transparency of the Universe to gamma rays. Our objective in this work was to use this physics to obtain bounds on a relativistic deformed kinematics, rather than focusing on the study of the transparency of the Universe. We acknowledge that the title may give the impression that the main point is the study of the Universe transparency, so we propose to change the title to "Bounds on Relativistic Deformed Kinematics from the physics of the Universe transparency".
Nevertheless, of course what is studied here has a lot of relevance for the general problem of the Universe transparency. This was briefly mentioned in the Introduction, with respect to the anomalies in the absorption of high-energy (refs. 28,29). To account for the request of the referee, we have added a paragraph in the conclusions which reads (when speaking of complementary studies):
"Consistency with these other observations will also indirectly contribute to a better knowledge about the physics of the transparency of the Universe to gamma rays, by constraining the role of new physics in the origin of possible anomalies in the detected gamma-ray spectrum or tracing them down to a lack of understanding of the EBL spectrum. The low-energy bounds obtained in the present work for a relativistic deformed kinematics makes this a promising approach with important astrophysical consequences."
Please note also that we have rewritten the second paragraph of the conclusions.
Reviewer 2 Report
The authors provide threshold equation for the process of gamma scattering, $\gamma\gamma \to e^+e^-$, in the framework of Relativistic Deformed Kinematics (RDK)
The authors have shown that the threshold condition (21) for RDK differs from the threshold condition (18) referred to Lorentz Invariance Violation (LV). The order-of-magnitude estimation for the bound on the energy scale $\Lambda_{eff}$ has been presented (eq. (23)). Surpisingly, this bound is many orders of magnitude lower than in the LV case. This is a very nice interesting result.
The authors wrote "We will have in near future new data on the very high-energy gamma-ray specrtum which could be used to get stringent bounds on the scale $\Lambda$". However, there is a lot of current data of this type from expirements H.E.S.S., HAWC, MAGIC, e.t.c. More detailed analysis can be provided with current data in a way similar to LV case, https://inspirehep.net/literature/1714057 https://inspirehep.net/literature/1701232
This possibility to apply the current data to more detailed analysis should be indicated in the second paragraph of the conclusion.
Author Response
We thank the referee for his/her comments and try to answer here to his/her main objection:
"The authors wrote "We will have in near future new data on the very high-energy gamma-ray spectrum which could be used to get stringent bounds on the scale $\Lambda$". However, there is a lot of current data of this type from experiments H.E.S.S., HAWC, MAGIC, e.t.c. More detailed analysis can be provided with current data in a way similar to LV case, https://inspirehep.net/literature/1714057 https://inspirehep.net/literature/1701232. This possibility to apply the current data to more detailed analysis should be indicated in the second paragraph of the conclusion."
The referee is certainly right. We have rewritten the second paragraph of the conclusion, which now reads:
"An analysis based on the modification of the threshold energy in the electron-positron pair production to consider the problem of the transparency of the Universe to high-energy gamma rays is of course incomplete and can only give qualitative indications. A more detailed analysis, taking into account specific models for the EBL and a determination of the optical depth, as in Refs. [19,26], should be performed. Such study will require to go beyond the determination of the threshold energy considering all the effects of the deformation of the kinematics in the determination of the VHE gamma-ray spectrum.
One could even consider a situation where the accuracy in the determination of the gamma-ray spectrum requires to go beyond the terms proportional to the inverse of the scale $\Lambda$ in the modification of the kinematics. There is at present a wealth of data from H.E.S.S., HAWC or MAGIC, where this analysis could be carried out, and we will have more data in the near future with CTA.
Such analysis can be used to get stringent bounds on the scale $\Lambda$ for a relativistic deformed kinematics if one does not find a conflict with special relativistic kinematics. Alternatively, one could find a spectrum which is not compatible with the predictions based on SR kinematics, and one should see whether the incompatibility could be adjusted with an appropriate choice of the scale $\Lambda$."
Please note also that two new sentences have also been added at the end of the third paragraph of the conclusions.
Reviewer 3 Report
The paper is devoted to investigation of the role of relativistic deformed kinematics in defining the threshold energy of electron-positron pair production in a photon-photon interaction. The authors were able to find the equation for this energy and they compare their result with the Lorentz invariance violation scenario. Of course the results obtained should be compared with the experiments on high-energy cosmic gamma rays.
I find the paper valuable and recommend it for publication as it stands.
Author Response
We thank the referee for his/her comments. We of course agree with the comment
"Of course the results obtained should be compared with the experiments on high-energy cosmic gamma rays."
This was explicitly stated in the final paragraph of the conclusions, but we have tried to give more relevance to this comparison with two new sentences at the end of the paragraph:
"Consistency with these other observations will also indirectly contribute to a better knowledge about the physics of the transparency of the Universe to gamma rays, by constraining the role of new physics in the origin of possible anomalies in the detected gamma-ray spectrum or tracing them down to a lack of understanding of the EBL spectrum. The low-energy bounds obtained in the present work for a relativistic deformed kinematics makes this a promising approach with important astrophysical consequences."
Please note also that we have rewritten the second paragraph of the conclusions.